# Water Consumption Range Prediction in Huelva's Households Using Classification and Regression Trees

**Gustavo Bermejo-Martín, Carlos Rodríguez-Monroy *** and **Yilsy M. Núñez-Guerrero**

Department of Industrial Organization, Business Administration and Statistics, E.T.S. Ingenieros Industriales, Universidad Politécnica de Madrid (UPM), Calle José Gutiérrez Abascal 2, 28006 Madrid, Spain; gustavo.bermejo.martin@gmail.com (G.B.-M.); ym.nunez@upm.es (Y.M.N.-G.)
* Correspondence: carlos.rodriguez@upm.es

**Abstract:** This paper uses the numerical results of surveys sent to Huelva's (Andalusia, Spain) households to determine the degree of knowledge they have about the urban water cycle, needs, values, and attitudes regarding water in an intermediary city with low water stress. In previous research, we achieved three different households' clusters. The first one grouped households with high knowledge of the integral water cycle and a positive attitude to smart devices at home. The second cluster described households with low knowledge of the integral water cycle and high sensitivity to price. The third one showed average knowledge and predisposition to have a closer relationship with the water company. This paper continues with this research line, applying Classification and Regression Trees (CART) to determine which hierarchy of variables/factors/independent components obtained from the surveys are the decisive ones to predict the range of household water consumption in Huelva. Positive attitudes towards improved cleaning habits for personal or household purposes are the highest hierarchy component to predict the water consumption range. Second in the hierarchy, the variable Knowledge Global Score about the integral urban water cycle, associated with water literacy, also contributes to predicting the water consumption range. Together with the three clusters obtained previously, these results will allow us to design water demand management strategies (WDM) fit for purpose that enable Huelva's households to use water more efficiently.

**Keywords:** CART algorithm; Design Thinking; web-based prototype; ICT technologies; water demand management (WDM); household; water consumption

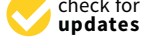



## 1. Introduction

This paper's relevance is that it is based on research done in a city like Huelva (around 140,000 inhabitants in Andalusia, Spain), presenting low water stress. As mentioned by Bermejo-Martín et al. (2020). "The absence of a sensation of scarcity does not act as a motivating lever among citizens to improve efficiency in water use. Without this motivating driver, the entire project's goal is to determine to what extent a more efficient and sustainable use of water can be achieved in households through strategies focused on water demand management (WDM) and not supply" [1].

On the other hand, we have revised the literature regarding cities' existing typology based on their size. We highlight that the research is carried out in an intermediary city [2] such as Huelva. Most of the studies with similar objectives are focused on large cities and megalopolis [3], which suffer from water scarcity. We consider it interesting to analyze households' starting situation in their relationship with water in an intermediate city to measure the extent to which it is possible to achieve citizen engagement in a context where there is no water crisis.

The citizens' needs about water, or water social needs, are embedded in a specific city context. In this sense, we have also reviewed the existing literature regarding concepts close to water social needs, such as habitability [4], sustainability [5], and hydrosocial contract [6]. We also highlight the importance of the water company, since water companies

are in charge of covering most of these social water needs. The relationship between water companies and citizens is mainly reflected in water consumption in the home, impacting the habitability [4] and sustainability [5] of the city. Given the impact of citizen behavior, it is necessary to implement some strategies to achieve the citizens' engagement in more efficient use of urban water. Thus, the water social needs covered by the services already deployed and the existing hydrosocial contract in the city must be taken into account.

"The hydrosocial contract is understood as the values and, often, implicit agreements between communities, governments, and businesses on how water can be used. This hydrosocial contract will reflect citizens' practice and behavior in water use within the city and the needs and mechanisms used to satisfy them," as appears in the previous paper of Bermejo-Martín et al. (2020) [6].

The main urban challenges that affect Huelva's water are climate change and unstable and unpredictable water social needs [7] with an economic growth need. Population in Huelva has high social diversity, but it is stagnant or with natural increase. This situation requires WDM strategies "fit for purpose" for each type of citizens groups to improve water efficiency use, according to Dean et al. [8].

Dean et al. [8] propose a methodological framework that has three pillars of the citizens' commitment to their city's water:

1. Understand water and its context. It could be understood as the knowledge about water.
2. Value water through emotions and the development of positive attitudes, such as:
   a. Support for alternative sources of water.
   b. Pro-environmental perspective: the environmental identity of the home.
3. Support the smartest daily behavior regarding water in different spheres,
   a. Public:
      i. Through the influence on political processes.
      ii. Support and pressure to repair damaged waterways.
   b. Private:
      i. By saving and using water efficiently.
      ii. Using water-saving devices.
      iii. Reducing water pollution.

If we want to improve citizen engagement, we must act on all three pillars. In this research, we only contemplate the sphere of private behavior of citizens within their homes.

In this sense, this research continues along the lines of the published article [1], where the authors showed the numerical results and the analysis of households' degree of knowledge, needs, values, and attitudes in an intermediary city such as Huelva, about sustainable urban water use. It was the first descriptive analysis and origin for subsequent development within the research. This paper mostly uses the scientific literature of other previous articles by the same authors [1,6], which are referenced in this introduction.

Based on those factors, knowledge, needs, values, and attitudes, we have obtained three clusters [9] of households based on high, medium, and low knowledge of the integral urban water cycle. The first cluster grouped households with high knowledge of the integral water cycle and a positive attitude to smart devices at home. The second one described households with low knowledge of the integral water cycle and high sensitivity to price. Finally, the third one showed average knowledge and predisposition to have a closer relationship with the water company. The clustering technique chosen was hierarchical, Ward's method [10,11]. This classification was the starting point to build a web-based prototype [12] within a Design Thinking (DT) methodology [13] as described in [6]. As we have already commented, we pretend to build different effective WDM strategies to fit each cluster's purpose through this tool. The WDM strategies are three: the application of technology, the use of price, and the promotion of changes in consumer behavior and habits [14].

This research aims to check the weight of the different WDM strategy proposals to achieve greater household engagement concerning water and induce them to use it more sustainably. The first WDM proposals will be based on results obtained from [1] conclusions and from these paper learnings.

The purpose of this paper is to rank the influence of the different variables and components/factors extracted from the knowledge, needs, values, and attitudes shown by households in Huelva [1] on their range of water consumption (high, medium, low). For this, we have used classification and regression trees applying the Classification and Regression Trees (CART) algorithm. The reasons for this choice are detailed later. Based on the results obtained by applying CART, we will be able to act with WDM "fit for purpose" strategies that improve water use efficiency at home. The dependent variable is the range of consumption.

Before choosing the CART algorithm, we have reviewed scientific literature on prediction algorithms, especially other frequently used regression tree algorithms different from CART [15]. Most tree algorithms follow similar steps and principles. At some point in the induction stage, the differences among them are the measurement method of node impurity, split variable or split point choice, or tree pruning. They can process different types of variables like online or dynamic data, longitudinal data, big sample sizes, substantive data models, and various error structures [16]. We can stand out among them the Chi-squared Automatic Interaction Detector (CHAID) (Kass, 1980) [17], C4.5 (Quinlan, 1993) [18] is an extension of the ID3 (Quinlan, 1986) [19], Fast and Accurate Classification Tree (FACT) (Loh and Vanichsetakul, 1988) [20], QUEST (Loh and Shih, 1997) [21], Classification Rule with Unbiased Interaction Selection and Estimation (CRUISE) (Kim and Loh, 2001) [22], Generalized, Unbiased, Interaction Detection, and Estimation (GUIDE) (Loh, 2009) [23] and Conditional Inference Trees (CTREE) (Hothorn et al., 2006b) [24].

The variables/factors/components were obtained by processing the responses to a survey with 28 questions sent to 97 Huelva's households in September 2018. Greater detail about them can be found in [1]. We followed the same methodology as Dean et al. [8] to design and process the survey answers.

This paper's novel contribution is having verified that the character of Huelva as an intermediary city and without a shortage of water conditions the choice of WDM strategies to act on more efficient use of water. This situation places the homes of Huelva, far from previous experiences of citizen engagement in large cities and megalopolis [3], with a water crisis. In these large cities, the main levers that influenced household behavior concerning water are social behavior, demographic characteristics, or the household area within the city [8]. In the case of Huelva, the proposed actions that take into account these factors would not make much sense.

The results of this research show that in Huelva, the primary factors in building successful WDM strategies that allow a more efficient consumption of water in homes must take into account:

1. the attitudes about "Efficient use in cleaning."
2. the components associated with water literacy. Fundamental is the household knowledge about the water cycle and Education variables.

## 2. Materials and Methods

This paper's origin was to respond to a challenge set by the UP4 Solutions program to get more efficient and sustainable use of water in homes. This program is managed by the Suez group and the Spanish UP4 technological universities alliance (Universitat Politècnica de Catalunya, Universidad Politécnica de Madrid (UPM), Universitat Politècnica de València, and Universidad Politécnica de Cartagena) [25].

"The main objective of the challenge is to promote the efficient and sustainable use of water resources based on information, communication, and awareness of citizens. The project focuses on understanding, within the water cycle, what are the factors, elements, and data that generate the greatest impact and understanding of the challenges faced in

terms of water resource management, thus being able to generate a reflection of households towards a cultural change regarding their use and valorization" [6]. Specific goals were defined in [1]:

1. Evaluate the degree of knowledge of consumers about the challenges associated with water's sustainable use.
2. Identify their needs and values as consumers.
3. Evaluate attitudes towards different aspects of water:

    a. Attitude towards consumption efficiency.
    b. Attitude towards the adoption of technological devices in the home related to water.
    c. Attitude towards reclaimed water in the home.

4. Determine prototype solutions that help raise awareness and value water use sustainability, proposing specific actions to consumers that address part of their needs at the same time.

The city of Huelva, in Andalucía (Spain), has approximately 140,000 inhabitants (intermediary city). Aguas de Huelva [26] is the company that operates the water network in the city and gives services to them. It is participated by the Suez group. Aguas de Huelva has around 52,000 clients, most of whom are households. Only 9350 of them receive a digital invoice, from which we can deduce that they are accustomed to using Information and Communication Technology (ICT) tools and has the proper technological infrastructure to communicate with them through digital channels" [6]. For research, logistics, and time costs, a sample of 350 households out of those 9350 were invited to participate in the project. These 9350 client households are the universe from which it was started, which presents sufficient homogeneity among its samples. The first step of the project was the preparation of a survey. The survey was launched via email with a link to questions to more than 350 households. After successive rounds of sending the survey to raise the number of participating households, the commitment of participation of 120 households was achieved. Finally, we got valid responses to the survey from 97 of them.

The number of participating households is one of the limitations of this research. It is extremely difficult to find participating households with diverse profiles to minimize the effect of uncertainty, the personal preferences of each household, the possible errors in obtaining household data, misunderstandings in language and communication with households, and inconsistencies, among others, as recognized in [27]. However, the number of participating households is in line with other similar investigations [28–33].

The partners of this project approved a plan based on DT methodology in July 2018. This methodology constitutes a general model or research framework. This paper and [1] form part of the needfinding and synthesis stages of DT defined in the Stanford ME310 model [13] we adopted. From 172 DT methods identified [34], we have chosen this one because it is the most used according to service companies' ranking [34]. DT has been applied in similar cities with smart water initiatives in Brazil [35] and South Africa [36]. DT has been the methodological framework for water management, water treatment and purification products, water management design after environmental disasters [37], and facing problems like floods and low water quality in Morocco [38].

To achieve all the goals, the project's planned time scope was set up for one year to cover Spain's natural water cycle's full seasonal effect. Due to logistics reasons, costs, and time, among others, we finally obtained a sample of 97 households to participate in this research.

We elaborated a 28-question survey for households to evaluate the three axes in 30 min in December 2018. Finally, in mid-June 2019, we closed the survey response collection after several rounds of survey submissions. Besides, we also reviewed the available scientific documentation and finally, we decided to apply the model by Dean et al. [8] because it seemed the most reliable and adapted to the research context. "This model follows a methodology based on the survey questions' design, the sending of the said survey, and

the mathematical analysis of the household survey data. Besides, this model also made it possible to cover the DT synthesis phase employing clustering groups of households based on needs valorization and similar behaviors" [1].

We sent the survey to households through email with a link to the Aguas de Huelva website where the survey is placed. The questions/answers followed a 5-level Likert scale, associating them with major factor blocks:

1. Demographic characteristics: such as age, gender, education, annual income, and professional occupation. For the purposes of this paper, we would like to highlight the independent qualitative variables:

    a. "Education", which measures the educational level of households, with the following possible values:

        i. No studies.
        ii. Basic studies.
        iii. Bachelor.
        iv. University studies.

    b. "Professional occupation", which measures the origin of income sources, with the following values:

        i. Unemployed.
        ii. Employed.
        iii. Self-Employed.
        iv. Retired.

    c. "Income", which measures the range of household income, with the following values:

        i. Less than 10,302 €/year.
        ii. From 10,302 € to 25,000 €/year.
        iii. From 25,000 € to 50,000 €/year
        iv. From 50,000 € to 100,000 €/year.

2. Home characteristics: Number of people in the household, children in the home, surface area, age.
3. Life experiences and psychosocial factors: Household activities around the city water, active participation in any social organization, and if they have suffered any water restriction.
4. Needs, values, and identity regarding water: The importance of different aspects of water, consumption concerning neighbors, identification of better use of water within the home, the possibility of the investment amount in water-saving devices, and acceptance of water regenerated in the home (support to alternative sources of water, basically regenerated rain water).
5. Knowledge of the home regarding the integral water cycle, ordered in the following blocks: uptake, treatment, distribution, sewerage networks, purification, and regeneration.

In the coding stage before data processing, we added to the response table the following fields provided by Aguas de Huelva:

1. Huelva area to which the home belongs: Six possible areas that correspond to the social districts that Huelva's City Council applies to the city.
2. Average consumption per person within the home: liters/person/day. They correspond to the average of the year 2018.
3. Consumption range of each household, according to this classification agreed with Aguas de Huelva:

    a. Less than 100 L/person/day is low consumption.
    b. Between 100 and 130 L/person/day is a medium consumption [34].
    c. More than 130 L/person/day is high consumption.

The consumption range has been built as a qualitative dependent variable from a quantitative variable, which is the consumption in liters per person and day in each household. It was set, considering that:

1.  The average consumption per inhabitant per day in households managed by Aguas de Huelva is 126.5 L/inhabitant/day.
2.  The minimum vital water consumption set by the World Health Organization is set to 50 L/inhabitant/day [39].

The knowledge range has been built as a qualitative dependent variable from a quantitative variable, which is the Knowledge Global Score. It was set considering that:

1.  Low knowledge range: Scores lower than 4 (minimum is 0 points).
2.  Medium knowledge range: Scores between 4 and 6.
3.  High knowledge range: Scores equal to or greater than 7 (maximum is 10 points).

Surveys have been statistically processed using IBM SPSS v. 25." [1].

Regarding the needs, values, and attitudes with respect to water, it was necessary to reduce the number of variables to facilitate their interpretation. For this purpose, a factor analysis by maximum likelihood with the principal components' extraction, with varimax with Kaiser normalization rotation method, was applied to each block.

The factors about what households value most were:

1.  Component 1 = Quality of service/health. This component explains what households value most. They are aspects related to water quality and its impact on health.
2.  Component 2 = Quality of infrastructure. Aspects related to the water network's infrastructure, such as breakdowns in the network, cuts in the water service in the home, low water pressure in the home, and flood management.
3.  Component 3 = Customer Relationship Management (CRM). It is the value of the household's relationship with the water company. It shows the importance of the service brand of Aguas de Huelva among households. It is significant because it is useful to characterize the trust households could have in the service company. It is the beginning to predict reaction to engagement issues proposed to them in the later stages of the project.

The factors evaluating the individuals' attitudes towards water consumption efficiency, towards the adoption of technological devices in the home related to water, and towards reclaimed water in the home were:

1.  Component 1 = General efficient use to improve the efficiency consume of water in the home. "There is an association of variables that would explain a general use to improve the efficiency of water in the home" [1]. Households show a positive attitude to improve water use in this field.
2.  Component 2 = Efficient use in cleaning for personal or household purposes. "A group associated with a field of improvement in water use in personal or household cleaning tasks" [1].
3.  Component 3 = Water saving devices.
4.  Component 4 = Reclaimed water for washing machine, toilet, garden, and no case.
5.  Component 5 = All previous cases at a lower price.

All these factors or components are described because they will be taken as independent variables to rank them and build the first WDM strategies.

We can highlight other research projects which follow similar methodology like [28] on environmental lifestyle, Ref. [29] on community knowledge about water, Ref. [30] mixing quantitative and qualitative research methods on end-use water consumption in households, Ref. [31] on the use of surveys and case studies of an Australian water utility, and Ref. [32] regarding individual household survey responses to predict household environmental outcomes.

In the former paper [1], we developed arguments to generate clusters focused on the end-use of household water consumption, such as [9]. We mentioned different issues about the cluster analysis technique that can be found in [10,11].

Concerning the methodology followed in this paper, we can say that the Classification and Regression Trees (CART) is an algorithm to build a model from data. CART and C4.5 [40] are possibly the most used and popular algorithms to classify data [41]. Most researchers associate automatically "classification and regression trees" methodologies with CART and its software [16]. CART was developed by Breiman, Freidman, Olshen, and Stone in 1984 [42], but it was not the first regression tree algorithm published, which was the Automatic Interaction Detection (AID) (Morgan and Sonquist, 1963) [43]. After, Theta Automatic Interaction Detection (THAID) (Messenger and Mandell, 1972) [44] followed developing this new area [15].

Tree algorithms have been developed for various applications. They can create decision rules to distinguish between clusters of observations and determine the class of new observations. It is also used to analyze and segment a large amount of data quickly to detect and select important variables and interactions. Furthermore, one of the main applications is to have a fit for purpose data-driven prediction machine, which displays results in a user-friendly way. This is one of their main advantages, which is easily adapted to any problem [16]. These are the main reasons for applying this methodology in this paper.

There are two types of trees: the classification tree and the regression tree. The classification tree is used when there is a need to classify the data. The regression tree is used when the dependent variable is continuous [45].

CART is a non-parametric tool of discriminant analysis. A CART tree is a binary decision tree whose basic principle is to build a tree by splitting a node in half repeatedly. After every split, two child nodes are obtained. In summary, the tree structure can be interpreted as a decision rule and feature selection. The CART algorithm could be used with all types of data, including both categorical and numeric.

The process of building a tree has four stages: building, stopping, pruning, and selection. The whole process is well described in [46]: "Tree construction begins with the root node that contains the whole learning samples. If data in the node are of mixing classes, that node must be split. The splitting strategy is that the algorithm will search for all possible variables and all possible values to find the best split so that data in child nodes are of maximum homogeneity, sometimes referred to as purity. The key idea of CART is recursive partitioning. CART's process begins by taking all data to consider all possible values of all variables for growing a tree. So, it will select one variable or value that produces the best separation in the target attribute. If the value in focus is lower than the value at the separate point, that value will be placed on the tree's left side. For the value greater than or equal to the value at the separate point, it will be sent to the right side of tree". This splitting process is repeated until the separate point where it is not possible to improve purity.

This best separate point is checked by the Gini index [46]: "Gini is a measure of impurity computed by counting the frequency of events that how often a randomly chosen data instance is wrongly labeled, given that that instance is to be randomly labeled based on the distribution of class labels [45]".

In the context of this research, we have tried to work with the most homogeneous divisions of the data. This is another reason for having used the CRT algorithm. To evaluate the tree structure's goodness, cross-validation was applied. Cross-validation subdivides the data and builds a tree with each subdivision until the final tree is obtained. To avoid overfitting the model, the pruning method was used. In addition, the sensitivity and specificity analysis with ROC curves has been carried out, validating the method.

CART has demonstrated good medical application results, for example, in efficient diagnosis based on patients' symptoms [47]. However, there are numerous other examples in this area [48,49]. In economics, CART has been applied to decide the best way to manage business plans [50], or for example, to predict the Initial Public Offering (IPO) method [44]. For water purposes related to environmental applications, this algorithm can help predict rainfall and groundwater level [51] and groundwater potential for sustainable

planning, irrigation, and town water supply purposes to achieve water demand goals [52]. In household investigations, it has been applied also [53].

## 3. Results

Results have been statistically achieved using IBM SPSS v. 25 from Huelva household surveys' responses. The average of total knowledge gives a score of 4.2577, which indicates a degree of knowledge of the integral water cycle close to five, that is, a degree of medium knowledge, but still deficient in some of the stages of the urban water cycle.

The influence of the different variables and components/factors described in the introduction to this paper and obtained in [1] on the household consumption ranges (high, medium, low) has been analyzed. For this, the CART algorithm with cross-validation has been used, which allows determining which independent variables/factors/components can hierarchically influence household water consumption ranges. We take the volume of household consumption as an expression of households' real behavior in water use. The qualitative dependent variable is the range of consumption, constructed from each household's volume of consumption.

We have followed the procedure to analyze the hierarchy of the different factors over the water consumption range by blocks of values and needs, attitudes, and knowledge about the integral water cycle. We have added these step by step, trying to measure the weight of the combined effect. Subsequently, the hierarchy of all of them has been analyzed together.

Following this plan, we begin by analyzing the influence of the components that measure what households value most concerning water. In other words, only the values and needs of households have been considered to try to predict whether it is a household with high, medium, or low consumption. We do not consider attitudes or knowledge:

1.  Component 1 = Quality of service/health.
2.  Component 2 = Quality of infrastructure.
3.  Component 3 = CRM.

In this case, the algorithm does not discriminate based on these factors, and, therefore, no tree is obtained. In other words, considering only households' needs and values, the range of household water consumption cannot be predicted. The rest of the factors would not need to be considered.

Now, we do not consider values/needs or knowledge, and we only analyze the components that measure attitudes about their influence on the ranges of household consumption:

1.  General efficient use.
2.  Efficient use in cleaning.
3.  Water-saving devices.
4.  Reclaimed water for washing machine, toilet, garden, and no case.
5.  All previous cases at a lower price.

The attitude component that would best discriminate to predict the future range of household consumption would be a positive attitude towards "Efficient use in cleaning."

If now, we jointly analyze values/needs and attitudes, through their components, we obtain the following hierarchy tree and importance table as shown in Figure 1:

We can apply reasoning for this tree's interpretation, similar to that followed in [54]. As can be seen in Figure 1, it is a regression tree that begins with node 0. This reflects the dependent variable, the range of water consumption in the household, in its original state, as it was created. Overall, 38.1% of the samples have a low consumption, 27.8% have a medium consumption, and 34% have a high consumption.

The CART algorithm performs the first partition through the component "Efficient use in cleaning". Node 1 is obtained, which contains 48.3% of households with a high consumption range to this category, causing this node not to branch any further. Node 2 is also obtained, which contains 45.6% of the households with a low consumption range.

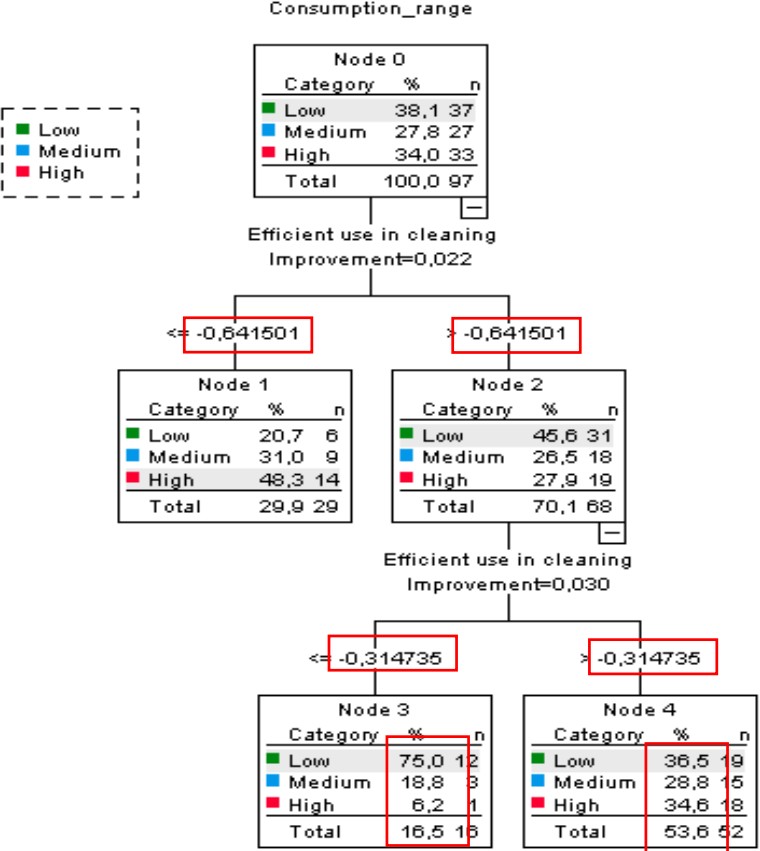

**Figure 1.** Summary tree of the Classification and Regression Trees (CART) algorithm on the joint influence of values/needs and attitudes to predict the range of household consumption. (Source: Authors).

From node 2, the algorithm performs another hierarchization, whose most influential component is again "Efficient use in cleaning". This means that this component will be the second most important within the group of independent variables studied. It also reinforces the importance of this component achieved in the first splitting. Nodes 3 and 4 are obtained. Both of them emphasize the category of low range consumption.

In conclusion, on the joint influence of values/needs and attitudes to predict the range of household consumption in Huelva, the hierarchy obtained reflects that it is the attitude component of "Efficient use in cleaning" that best predicts, followed in second place again for the same component "Efficient use in cleaning". The importance of each component can be summarized in Table 1.

**Table 1.** Importance of the components of values/needs and attitudes together to predict the range of household consumption (Source: Authors).

| Independent Variable | Normalized Importance |
|---|---|
| Efficient use in cleaning | 100.0% |
| Quality of infrastructure | 15.6% |
| CRM | 13.1% |
| Reclaimed water for washing machine, toilet, garden, and no case | 11.5% |
| Quality of service/health | 10.3% |

Attitude discriminates better than values/needs regarding predicting the future of the household consumption range.

Finally, if we jointly analyze the knowledge about the integral water cycle, values/needs, and attitudes of households, we obtain these results regarding the range of consumption (Figure 2):

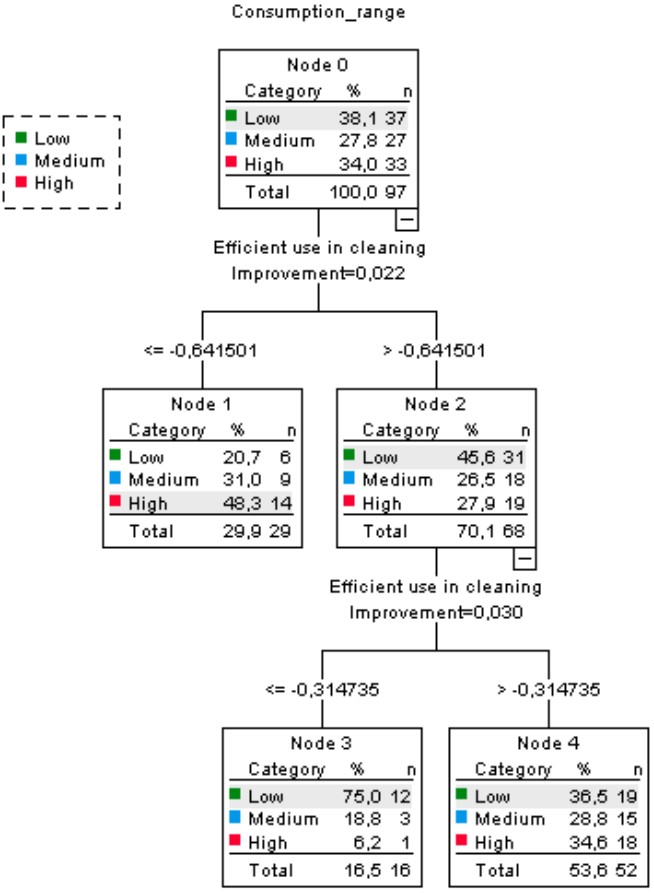

**Figure 2.** Summary tree of the CART method on the joint influence of knowledge, values/needs, and attitudes to predict the range of household consumption. (Source: Authors).

To characterize households' knowledge regarding the integral water cycle, we have taken the quantitative independent variable, the Knowledge Global Score. As we can see in Figure 2, the tree is the same as Figure 1. So, we can conclude that a positive attitude towards improving cleaning habits is the factor in predicting the water consumption range in Huelva households. What is different is the weight of importance of factors and variables. Table 2 presents the importance of these variables and components.

**Table 2.** Importance of the independent variable of knowledge, values/needs, and attitudes together to predict the range of household consumption. (Source: Authors).

| Independent Variable | Normalized Importance |
|---|---|
| Efficient use in cleaning | 100.0% |
| Knowledge Global Score | 33.7% |
| Quality of infrastructure | 15.6% |
| CRM | 13.1% |
| Reclaimed water for washing machine, toilet, garden, and no case | 11.5% |
| Quality of service/health | 10.3% |

Analyzing the importance of the variables and factors, we can see that knowledge about the integral water cycle appears as the second variable, with normalized importance of 33.7%. This suggests that we also focus on educating households (literacy) on aspects of the water cycle. In this sense, to confirm the weight of water literacy, we have repeated the previous analysis, but adding the educational level, the independent qualitative variable "Education", and the professional occupation, the independent qualitative variable "Professional occupation" to the variables and factors that we already had, and that were the knowledge about the integral water cycle, values and needs, and attitudes of households. The tree obtained is the same as in Figure 2, being the factor "Efficient use in cleaning" that predicts the range of household consumption. However, now analyzing the importance of these variables and factors, we can see in Table 3 that the educational level's weight reinforces the variable of knowledge about the integral water cycle. This indicates the importance of the aspects associated with the "literacy" about water:

**Table 3.** Importance of the independent variable of education, professional occupation, knowledge, values/needs, and attitudes together to predict the range of household consumption. (Source: Authors).

| Independent Variable | Normalized Importance |
|---|---|
| Efficient use in cleaning | 100.0% |
| Knowledge Global Score | 33.7% |
| Education | 26.6% |
| Quality of infrastructure | 15.6% |
| CRM | 13.1% |
| Reclaimed water for washing machine, toilet, garden, and no case | 11.5% |
| Quality of service/health | 10.3% |

Along the same lines, the previous analysis has been repeated, but suppressing the variable "Professional occupation" (it does not appear in Table 3 of importance), and adding the range of household income, the independent qualitative variable "Income," to the variables and factors that we already had. These were education, knowledge about the integral water cycle, values/needs, and households' attitudes. The tree obtained is the same as in Figure 2, being the factor "Efficient use in cleaning" that predicts the range of household consumption. Analyzing the importance of these variables and factors, it is repeated with the same values as in Table 3 that the weight of the educational level reinforces the variable of knowledge about the integral water cycle.

In summary, taking together the knowledge about the integral water cycle, the values/needs, and households' attitudes, the variable that best discriminates the range of water consumption is the positive attitude towards improving cleaning habits. However, we must consider the importance of the range of consumption, which represents the knowledge of the integral water cycle households in Huelva have.

## 4. Discussion

As points of discussion and interpretation of the entire analysis of the variables and components/factors that influence the prediction of the ranges of household consumption in the city of Huelva, we can deduce that:

1. Positive attitude towards improving habits in cleaning for personal or household purposes explains and predicts the range of household consumption. Strategies that will enhance efficiency in this regard, especially those that refer to water use in Bathtub and Shower, Washing Machine, and Dishwasher (component 2) [1].

2. We must emphasize as a point of discussion that the positive attitude towards improving habits reflects what households think about what they themselves can improve in their use of water. This is what the answers they give in the survey reflect. They describe the idea of their awareness of actions to improve, and therefore, it can be expected that

they will accept proposals and advice along these lines. It is striking that for households in Huelva, toilet flushing is secondary in explaining the water consumption efficiency, whereas it is a crucial source of water consumption in households. Toilet flushing is included in the "Toilet" variable of component 1 (Toilet, Household cleaning, Cooking). From our point of view, this fact means that households are not aware that they can improve water consumption if they use strategies or devices that regulate or reduce toilet flushing. It may be a point of discussion to set educational and WDM strategies that raise awareness of the importance of toilet flushing to homes.

3. Second in the hierarchy would be the components associated with water literacy. Especially important are the Knowledge Global Score and Education variables, as can be seen in Table 3. Strategies are advised to increase household knowledge about the water cycle and its efficient use.

4. The rest of the components and variables that characterize the rest of the attitudes and values/needs declared by the households in Huelva have a similar hierarchy and importance to predict the range of household consumption, as can be seen in Table 3.

5. It is necessary to emphasize that the characteristics of households' social behavior (belonging to social organizations, social activities) have no weight in the hierarchy over their range of consumption. Actions that enhance or use these levers are not recommended.

6. There is no weight on consumption by district or any other demographic variable (homogeneity of households in Huelva). Differentiated strategies by districts in Huelva or by any other demographic characteristic of the household would not make sense. It is an intermediary city, not a megalopolis, where the city's smaller size and population facilitate its homogeneity.

Because of the statistical analysis of the results obtained, the drivers that can best act on more efficient use of water, translated into a reduction in the consumption by the home, are WDM information strategies on the urban water integral cycle to change habits (by improving knowledge and literacy) and strategies on how to improve water use efficiency in Bathtub and Shower, Washing Machine, and Dishwasher. These strategies can be in the advising line, that is, again in the educational line, and in the line of the WDM strategy of technological improvement of washing machines and dishwashers' consumption. The positive attitude towards using reclaimed water for washing machine, toilet, and garden reflected in the normalized importance of this component that appears in Table 3, would also help propose strategies to households in the adoption of technological solutions that facilitate them.

In other studies carried out around the world, other levers influenced household behavior concerning water, such as social behavior, demographic characteristics, or the household area within the city [8]. However, this does not seem to be the case in Huelva. Therefore, it can be stated that the proposed actions that explore these lines would not make much sense.

A priori, it would not make much sense to use or deploy an ICT infrastructure with the primary objective of regulating the range of consumption of households in Huelva. This ICT structure is the one that would allow Advanced Metering Infrastructures (AMI) to be supported, that is, the installation of smart meters in homes. However, the analysis using the CART algorithm applied in Huelva does not reflect its deployment's relevance to support WDM policies. Attitude towards the adoption of technological devices in the home related to water, declared by households, represented by its component of "Water-saving devices," does not appear in Table 3 of importance on predicting the range of water consumption.

Furthermore, in the case of the company Aguas de Huelva, it currently does not have an extensive infrastructure of smart meters deployed throughout the city. In our case, out of the 97 homes in the study, only 9 have smart meters. If we wanted to implement policies based on them directed towards the home, we would have to make a big investment. A reference to the investment to be made for a massive deployment of Smart meters in another intermediary Spanish city such as Alicante can be found in [55].

Regarding saving water through the use of smart meters, numerous experiences carried out to date, such as the articles by Boyle et al. [56], Gangale et al. [57], Giurco et al. [58], Krishnamurti et al. [59], measure its impact on household water consumption habits. They collect water-saving results in a range that ranges from just 3% to 53%, which gives a considerable variability, depending on the conditions of the investigations carried out. Sønderlund et al. [60] provide an average saving of 20%. These studies suggest a high dependence on each city's context, making it challenging to take a savings figure as a reference. Given the disparity of results, we found no compelling arguments that recommend their deployment to improve water use efficiency in the home compared to the cost they represent.

In addition to the central role of smart meters as the primary technological tool, a central element in the digitization of the water network [55], it is necessary to highlight the possible impact on the use of water in the home that other smart devices such as visual displays with alarm (alarming visual display monitors). These can be installed in the shower, as in taps and other "internal sources of the home" of water.

Studies such as the one prepared by Willis [61] show that the savings achieved through this type of smart device can be around 15.4 L per shower. Considering an average of 2.65 showers per household per day, average savings of about 40.85 L/household/day are obtained, that is, about 27% of the water.

The cost of a device of this type, installation included, can be around 110 euros per device. The payback time of the investment in them depends on the prices of water and energy, but it can be less than two years. In practice, it also depends on the contextual use, more especially of the shower's duration, of the number of people per household, and effectively of the water price, all of these parameters being contextually dependent.

In the case of having the deployment of water-saving devices and consumption indicators in the home, combined with smart meters, and considering the research of Lee et al. [14], one might expect to obtain savings results above the mean estimated by Sønderlund et al. [60], that is to say, above 20%. This joint solution does seem to ensure a high saving impact, but it would be based on doubling the investment, which would make it difficult to implement in all households massively.

Nevertheless, the attitude towards the adoption of technological devices in the home related to water, declared by Huelva's households, represented by its component of "Water-saving devices," does not appear in Table 3 of importance on predicting the range of water consumption.

Therefore, the economic reasons for the investment to be carried out, and the degree of acceptance of smart devices by households in Huelva advise against their installation and deployment.

It is also expected to obtain better results of commitment from citizens, using information strategies, through prizes to achieve the household's savings objective. This would demonstrate the utility of combining purely educational strategies with incentive strategies, as demonstrated by Anda et al. [62].

Since there is no direct cause-effect relationship between attitude and real behavior [63], from experiments on using the web-based prototype, the limits for changing habits could be determined in later research phases. Without changing or detaching them from their daily home life context, define their limits to achieve greater efficiency in water use.

Both due to the results of the analysis using the CART algorithm shown in this paper, as well as the arguments set out above, before deciding on the deployment of Advanced Metering Infrastructures (AMI) and smart devices in the home, it would be convenient to try other strategies based on education, whose cost, efficiency, and speed of commissioning carry less risk and are cheaper.

In this sense, DT methodologies allow rapid and economic development of "web-based prototypes" based on a human-centered methodology [64,65]. These favor the engagement of households and mitigate the risk of development and adoption. Although according to Kumar et al. (2016) [64], "Design Thinking does not mitigate all the risk of

innovation. At times it may appear to make innovation riskier. Locating restless users for the cooperating user pool may not be easy; however, they are extremely helpful. Regular users are needed, as well. A challenge is to find a diverse set of restless users ".

This prototype as an initial step would allow a posteriori to scale the experience to all households in Huelva, minimizing risk and cost, as mentioned in a previous paper in Bermejo-Martín et al. (2020) [6] "using different ICT solutions, such as Big Data techniques and inference of future behaviors of homes through Artificial Intelligence" [6].

Besides, through this prototype, traceability could be carried out (using tools such as Mailchimp [66] and Google Analytics [67]) on the impact of use and adoption of this tool as an educational and knowledge channel of the integral urban water cycle of households. It can also be used to advise on how to improve water use efficiency in Bathtub and Shower, Washing Machine, and Dishwasher, especially in the purchase of washing machines and dishwashers with better water and electricity consumption. The education line focused on new technologies in the home that facilitate reclaimed water for washing machines, toilets, and gardens should be highlighted.

It should be noted that there is a temporary challenge in this type of research since most academic studies on strategies that explore the variation in household behavior take an average period of around a year, ranging from a minimum of six months to two years. In this way, it has been possible to verify the impact of the actions on the household's commitment to efficient water use behavior. Less time does not allow us to conclude anything about these strategies' effectiveness beyond a very short-term impact that is not explanatory.

Finally, given the abundance of tree algorithms, choosing the correct one is often a difficult task. There is a gap in a clear guide to be able to choose with guarantees. In our case, we have adopted CART because of its availability in commercial software packages such as IBM SPSS v. 25. Additionally, for its reliability and robustness demonstrated in numerous study applications described above.

Perhaps, as a future research line, a benchmarking could be carried out between the different classification and regression trees algorithms on the available data to check if there is a disparity in results, as has been done in other cases [15]. However, this makes perfect sense when the number of variables is very large, and choosing some preliminary variable selection can drastically improve the prediction. Along these lines, another possibility is to compare the results obtained by the CART algorithm with other machine learning tools, such as neural networks and random forest.

**Author Contributions:** All authors have participated directly in this research. Conceptualization, G.B.-M. and C.R.-M.; data curation, G.B.-M. and Y.M.N.-G.; formal analysis, G.B.-M. and Y.M.N.-G.; investigation, G.B.-M., C.R.-M., and Y.M.N.-G.; supervision, G.B.-M. and C.R.-M.; writing—original draft, G.B.-M. and C.R.-M.; writing—review and editing, G.B.-M., C.R.-M., and Y.M.N.-G. All authors have read and agreed to the published version of the manuscript.

**Funding:** This research received no external funding.

**Institutional Review Board Statement:** Not applicable.

**Informed Consent Statement:** Not applicable.

**Data Availability Statement:** Data sharing not applicable.

**Acknowledgments:** This study was partly supported by the Suez group, Aguas de Barcelona, and Aguas de Huelva.

**Conflicts of Interest:** The authors declare that they have no conflict of interest.

## Abbreviations

| | |
|---|---|
| AMI | Advanced Metering Infrastructures. |
| CART | Classification and Regression Trees |
| CRM | Customer Relationship Management |
| DT | Design Thinking |
| ETSII | Escuela Técnica Superior de Ingenieros Industriales |
| ICT | Information and Communication Technology |
| IPO | Initial Public Offering |
| LDA | Linear Discriminant Analysis |
| SDG | Sustainable Development Goals |
| UP4 | Network of Spanish Technical Universities |
| UPM | Universidad Politécnica de Madrid |
| WDM | Water Demand Management |

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
