# Peer review of "Water Consumption Range Prediction in Huelva’s Households Using Classification and Regression Trees"

_water, doi:10.3390/w13040506_

Round 1
Reviewer 1 Report
Hi
Thanks for submitting your paper. I put my notes in the attached file.
I also have some major comments here?
1- why did you use very old algorithm and methodology? There are several new decision tree models that can provide more accurate results. you can read one of the samples here: https://ascelibrary.org/doi/abs/10.1061/9780784483206.010
2- there are forward and backward variable selection methods to select the appropriate variables. why did you do that manually?
3- how did you train and test your model? did you use all the data to train the algorithm? if yes how did you test it?
4- decision tree models are very likely to be overfitted. did you use any method to protect the results?
that would be great if you can change your model or clear your training process.
Thanks

Reviewer 2 Report
The manuscript "Water Consumption Range Prediction in Huelva’s Households using Classification and Regression Trees" using numerical results of surveys sent to Huelva households (Andalusia, Spain) to determine the level of knowledge about the urban water cycle, needs, values and attitudes to water in the intermediate city with low water stress. The CART methodology is well explained at the expense of the statistical software IBM SPSS v. 25, with which the authors also work when processing the results. Nevertheless, in order to fully understand the submitted manuscript, it is necessary for the reader to read the previous publication of the authors mentioned in the References [1].
Keywords contain words identical to the title of the article, what is undesirable (Classification and regression trees, households, water consumption range).
Another question is the fact that only 97 clients out of 52,000 Aguas de Huelva enter the results. Can such a low number of respondents be sufficient for the next presentation of results?
Lines 281 to 303, which contain a description of other regression tree algorithms used and subsequently not used in manuscript, are unnecessary in the Materials and Methods chapter.
Chapter results: In the description of tables 1,2,3 the authors describe the importance of factors and variables entering into the calculation, but the values of Importance and Normalized Importance are not sufficiently explained. Only the value is given here: eg 33.7%. These numbers are only constants of the authors they have listed in the tables.
The discussion is set up as a statement of fact in several points. Tab 4 mentioned in the discussion coincides with the already published Tab 3 in the manuscript [1] mentioned in the results chapter. Below are "ideas" that could be successful in reducing water consumption. The discussion also includes the financial balance sheet of Aguas de Alicante, which is no longer targeted directly at the respondents. Similarly, respondents' considerations of water savings using alarming visual monitors and similar devices are not part of the research results presented by the authors. Therefore, this information is irrelevant for the manuscript. This part of the paper lacks a comparison of results from other cities in order for these outputs to fulfill its research character.
There is no absence of the Conclusion chapter in the manuscript.
I do not recommend Manuscript in the form presented in this way for publication.
Reviewer 3 Report
Keywords: Six of the 9 keywords are linked with the methodological approach, whereas there should be sufficient number of keywords on the topic, that seems not to be evident considering the global set of keywords. I suggest equilibrating the set of keywords.
Global comments on the paper:
The paper is an assessment of water consumption prediction in household of a Spanish city. This subject is a key issue considering the necessity to limit the household extra consumption in cities in water stress context. Meanwhile, the paper needs some improvements before publication. You’ll find hereafter my comments/proposal/questions:
- Comment 1: Line 33: Provide the size of the city the first time you present the city.
- Comment 2: Lines 30-100: The introduction lacks introducing the issue. Most of this introduction is a part of the methodology, not a real introduction. There is no presentation the general trend of population use/habits/drivers regarding water, nor of other existing publications on that, and in what way the current paper adds new insight on that subject. I suggest reshaping that part of the paper, transferring a part of these lines in the methodology, and add global contextual approach in the introduction.
- Comment 3: Lines 152-153: 97 samples is low for statistical purpose. How can you be sure to be conclusive.
- Comment 4: Lines 189-190: What do you mean by “city water”? Could you clarify?
- Comment 5: Lines 195: Maybe you should precise what are the existing alternative sources of water in these Juelva's homes.
- Comment 6: Lines 215-216: The true value is 50, not 100 (that is in contradiction with reality). Therefore, you cannot indicate a range from 50 to 100. I suggest you indicate “50”.
- Comment 7: Line 440-441: What about toilet flushing? How could you explain that this use is secondary in explaining the water consumption efficiency, whereas it is a key source of water consumption in households?
- Comment 8: Lines 470-472: How can you be sure that this attitude is similar for use of reclaimed water for toilet/garden and for washing machine. Practically I anticipated that people would more easily have positive attitude for toilet flushing or garden than for washing machine. Did you specifically differentiate these uses in your study?
- Comment 9: Lines 475-476: It seems to be not supported by your results, as no linkage have been found with demographic variable, with district, nor I imagine with income.
- Comment 10: Lines 530-537: In practice, it also depends of the contextual use, more especially of the duration of the shower, of the number of people per household and effectively of the water price, all of these parameters being contextually dependent.
- Comment 11: Lines 576-577: My personal point of view is that this is evident and easy, as it is frequently used as a sale commercial argument. There is more need to advise optimization of the use the washing machines and dishwasher, that is not necessarily a current practice.
- Comment 12: Lines 580-586: I agree with you, it is at minimum necessarily to have a one-year survey.
Round 2
Reviewer 1 Report
.
Author Response
There are no comments or suggestions from the reviewer in the revision's second round.
Reviewer 2 Report
The authors responded to almost all the comments of my review. After editing the text, I recommend the manuscript for publishing.
Author Response
There are no comments or suggestions for authors in Revision Round 2.
Previous comments have been incorporated.
Reviewer 3 Report
Thanks for the explanations and amended paper. I have two remaining comments.
- Keywords: You have suppressed words that are on the topic I suggested you boost. You should keep "household" and “water consumption" that are not methodological words. Conversely, I suggest suppressing "engagement" that is an indistinct word.
- Lines 32-36 and 52-55 If these sentences are from other papers as it seems to be the case, you should introduce the sentences with something such as "As mentioned by ... et al (year) ...."
Author Response
In Round 2 there are no comments or suggestions from Reviewer 3. Previous comments and responses to both Academic Editors have been specified in the new Cover Letter.